# META-K: TOWARDS SELF-SUPERVISED PREDICTION OF NUMBER OF CLUSTERS

## ABSTRACT

Data clustering is a well-known unsupervised learning approach. Despite the recent advances in clustering using deep neural networks, determining the number of clusters without any information about the given dataset remains an existing problem. There have been classical approaches based on data statistics that require the manual analysis of a data scientist to calculate the probable number of clusters in a dataset. In this work, we propose a new method for unsupervised prediction of the number of clusters in a dataset given only the data without any labels. We evaluate our method extensively on randomly generated datasets using the scikit-learn package and multiple computer vision datasets and show that our method is able to determine the number of classes in a dataset effectively without any supervision.

## 1 INTRODUCTION

Clustering is an important task in machine learning, and it has a wide range of applications (Lung et al. (2004); Aminzadeh & Chatterjee (1984); Gan et al. (2007)). Clustering often consists of two steps: the feature extraction step and the clustering step. There have been numerous works on clustering (Xu & Tian (2015)), and among the proposed algorithms, K-Means (Bock (2007)) is renowned for its simplicity and performance. Despite its popularity, K-Means has several shortcomings discussed in (Ortega et al. (2009); Shibao & Keyun (2007)). In particular, with an increase in the dimensionality of the input data, K-Means' performance decreases (Prabhu & Anbazhagan (2011)). This phenomenon is called *the curse of dimensionality* (Bellman (2015)). Dimensionality reduction and feature transformation methods have been used to minimize this effect. These methods map the original data into a new feature space, in which the new data-points are easier to be separated and clustered (Min et al. (2018)). Some examples of existing data transformation methods are: PCA (Wold et al. (1987)), kernel methods (Hofmann et al. (2008)) and spectral methods (Ng et al. (2002)). Although these methods are effective, a highly complex latent structure of data can still challenge them ( (Saul et al., 2006; Min et al., 2018)). Due to the recent enhancements in deep neural networks ( Liu et al. (2017)) and because of their inherent property of non-linear transformations, these architectures have the potential to replace classical dimensionality reduction methods.

In the research field of *deep clustering*, popularized by the seminal paper *"Unsupervised Deep Embedding for Clustering Analysis"* (Xie et al. (2016)), deep neural networks are adopted as the feature extractor and are combined with a clustering algorithm to perform the clustering task. A unique loss function is defined which updates the model. Deep clustering methods typically take $k$, the number of clusters, as a hyper-parameter. In real-world scenarios, where datasets are not labeled, assigning a wrong value to this parameter can reduce the overall accuracy of the model. Meta-learning, a framework that allows a model to use information from its past tasks to learn a new task quickly or with little data, has been adopted by a handful of papers (Ferrari & de Castro (2012); Ferrari & De Castro (2015); Garg & Kalai (2018); Kim et al. (2019); Jiang & Verma (2019)) to improve the performance of clustering tasks. Closest to our work is the approach proposed by Garg & Kalai (2018) that tries to predict the number of clusters in K-Means clustering using meta-information.

To solve the same issue, we propose Meta-k, a gradient-based method for finding the optimal number of clusters and an attempt to have a self-supervised approach for clustering. Our work is based on the observation that a network can take input points and learn parameters to predict the best number

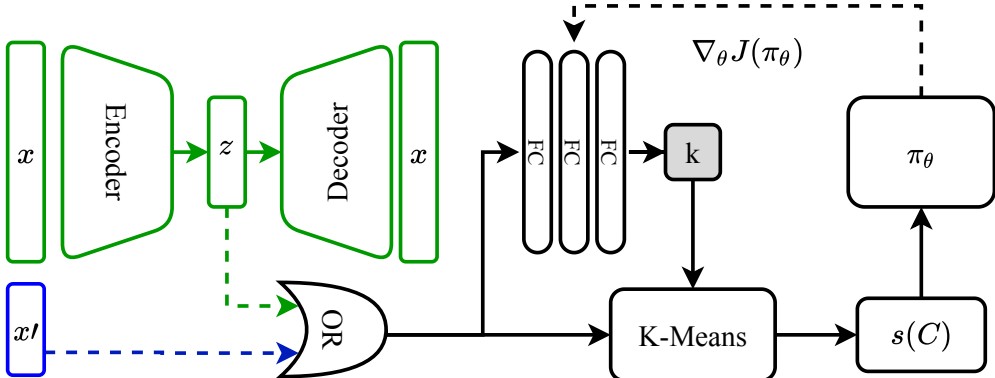

Figure 1: This figure shows an outline of our method. The green and blue dashed lines show two possible inputs to the clustering. The green part shows the high dimensional input (e.g. images) feature extraction using an auto-encoder and the blue line shows the low dimensional input. The right part of the figure shows the prediction of $k$ and how we update the controller network.

of clusters, $k$. The predicted $k$, along with the points in the dataset, are the inputs to the clustering algorithm K-Means. For the clusters created, the silhouette score (Rousseeuw (1987)) is calculated. Using this metric value as the reward signal, we can compute the policy gradient to update the controller. As a result, in the next iteration, the controller gives higher probabilities to the $k$ that causes better (closer to 1) silhouette scores to be calculated.

To be able to perform optimized clustering on both low- and high-dimensional spaces, we augment our model with a feature extraction module, a deep auto-encoder. In this regard, our work is related to the idea of *learning to learn* or *meta-learning*, a general framework to use the information that is learned in one task for improving a future task. Figure 1 shows the diagram of our model for low- and high-dimensional data. We evaluate our method in multiple scenarios on different clustering tasks using synthetic and computer vision datasets, and we show that our approach can predict the number of clusters in most settings with an insignificant error.

Our contributions are:

- A novel self-supervised approach for predicting the number of clusters using policy gradient methods.

- Extensive evaluation on synthetic scikit-learn ( Pedregosa et al. (2011)) datasets and well-known vision datasets MNIST ( LeCun et al. (2010)) and Fashion-MNIST ( Xiao et al. (2017)).

- Our results show that our approach is able to predict the number of clusters in most scenarios identical or very close to the real number of data clusters.

- We plan to release the source code of this work upon its acceptance.

## 2 RELATED WORK

There is a vast amount of unlabeled data in many scientific fields that can be used for training neural networks. Unsupervised learning makes use of these data, and clustering is one of the most important tasks in unsupervised learning. For unsupervised clustering, the classical K-means algorithm (Lloyd (1982)) has been used extensively due to its simplicity and effectiveness. If the data is distributed compactly around distinctive centroids, then the K-Means algorithm works well, but in real life, such scenarios are rare. Therefore, research has focused on transforming the data into a lower-dimensional space in which K-Means can perform successfully. If our data points are small images, then PCA (Wold et al. (1987)) is commonly used for this transformation. Other methods include non-linear transformation such as kernel methods (Hofmann et al. (2008)) and spectral methods (Ng et al. (2002)).

Deep learning has also been used for unsupervised clustering specifically because it can process high dimensional data effectively by learning embedding spaces (Schmidhuber (2015)). Deep clustering includes two phases: the feature extraction phase and the clustering phase. Although the extracted features can be fed directly to standard clustering algorithms, deep learning models usually optimize further over specific clustering losses. Xie et al. (2016) use a loss, based on student t-distribution and can accommodate for soft clustering. They train a stacked denoising auto-encoder as their feature extraction architecture. Caron et al. (2018) is an unsupervised training method that employs pseudo labels generated by the K-Means algorithm applied to the output of a convolutional network for the task of classification. Even though (Xie et al., 2016; Caron et al., 2018) achieve promising results, the number of clusters in K-Means and the classifier architecture remain as hyper-parameters needed to be tuned.

To optimize the task of clustering, we can leverage the meta-learning framework. According to Vanschoren (2018), "Meta-learning, or learning to learn, is the science of systematically observing how different machine learning approaches perform on a wide range of learning tasks, and then learning from this experience, or meta-data, to learn new tasks much faster than otherwise possible." Meta-learning is closely related to one-shot or few-shot learning, and it has shown promising results in supervised learning tasks. When data is limited or quick weight optimization is a must, meta-learning can benefit standard classification and also challenges in the area of Reinforcement Learning.

To our knowledge, there are only a few works that focus on using a meta-learning framework for unsupervised learning tasks. Ferrari & de Castro (2012); Ferrari & De Castro (2015) estimate which of the pre-existing clustering algorithms works well for a new clustering task. Their approach is limited to the algorithms that the user must provide and the number of clusters in K-Means is a hyper-parameter. Garg & Kalai (2018) Focuses on the theoretical foundations for meta-clustering and uses meta-attributes for learning the best number of clusters. The 399 datasets provided for training a binary similarity function are all labeled, and this means that during training, the number of clusters for each dataset is known. Kim et al. (2019) proposes a novel meta-learner called MLC-Net that mimics numerous clustering tasks during the training to learn an effective embedding space for new clustering tasks. In this regard, their work has a resemblance to metric-based meta-learning algorithms, and learning the best number of clusters is out of their focus. In Jiang & Verma (2019), a framework is introduced, which finds the cluster structure directly without having to choose a specific cluster loss for each clustering problem. They propose a stacked-LSTM (Hochreiter & Schmidhuber (1997)) as their architecture, and the number of clusters is a hyper-parameter of this network.

## 3 META-K

Our proposed method, **Meta-K**, is an attempt to self-supervise the unsupervised clustering task by automating the process of finding the best number of clusters, $k$. We adopted the meta-learning framework and the policy gradient algorithm in Meta-K. Our pipeline has two phases, training and inference. During the training phase, given we have high dimensional input $x$, we train our feature extractor $\phi(x)$ using reconstruction loss as shown in Equation 1:

$$\mathcal{L}(x, \hat{x}) = \|x - \hat{x}\|_2 \tag{1}$$

When the auto-encoder is fully-trained, a low dimensional latent representation $z$ is learned by the model. It is shown in previous research that the K-Means algorithm, in particular, performs better when the inputs are of lower dimensions; therefore, we use this representation as input to the clustering algorithm.

The extracted features from the encoder ($z$) or low dimensional input $x\prime$ are fed to both the K-Means algorithm (with a randomly sampled $k$ value from a predefined range with Gaussian distribution) and the controller network $\theta(x\prime)$ which has the job of predicting $k$. The controller network consists of multiple fully connected layers with a final layer, with fixed output length that shows the range of the $k$ values predicted by the controller. The length of this vector is denoted by $n$. We update the parameters of $\theta(x\prime)$ using the policy gradient method for the chosen $k$ value. In a standard reinforcement learning setting, the gradient is considered 1 when a certain action is taken and it

is set to 0 for the other actions. In our pipeline, we use the value of silhouette score $s(C)$ for the gradients. In other words, the goal of the controller is to find the value of $k$ which gives us the highest silhouette score. Silhouette score is a metric to evaluate the quality of clusters when we do not have access to the labels. Equation 2 shows the calculation of silhouette score.

$$s(C) = \frac{1}{|\cup C|} \sum_{x \in \cup C} \frac{b(x) - a(x)}{max\{a(x), b(x)\}} \tag{2}$$

Where $b$ is the mean distance to the points in the nearest cluster, and $a$ is the mean intra-cluster distance to all the points. $C$ is the clustering output from K-Means and $x$ is each data point belonging to an assigned cluster. The output of $s(C)$ is in the range of $(-1, 1)$ with 1 showing the perfect clustering. During the inference phase, the inputs are fed to the controller. The number of clusters is predicted by the network, and K-means clustering is performed using the predicted $k$ value. Figure 1 shows the outline of our method. Over the next sections, we explain our approach in detail.

### 3.1 POLICY GRADIENT OPTIMIZATION

To try to map our model to the reinforcement learning (RL) framework, we can look at the batch of inputs at iteration $t$ as observations $\mathcal{O}_t$, the controller as the policy network $\pi(a_t)$ and its output as the distribution of actions $a_t \in \mathcal{A}$. We need to randomly sample an action $a_t$ from the action space $\mathcal{A}$ and perform that action on the environment. Here we use the sampled value as the input $k$ for the K-Means algorithm. K-Means gets two inputs, $k$ and the batch of input features $x\prime$. The output is the clusters $(C)$ created by this algorithm. By calculating the silhouette score $s(C)$, we compute our reward signal $r_t \in R(a)$. In our problem, we think of silhouette score as the reward function as it needs to be maximized. If our controller has learned the optimum number of clusters, then the K-Means algorithm builds clusters that are well separated and cohesive. Therefore, silhouette score is maximized. The controller (policy network) has the goal of maximizing the cumulative discounted reward $R_t = \sum_{k=0}^{\infty} \gamma^k r_{t+k}$ with the discount factor $\gamma(0, 1]$ in one or more trajectories. In RL terminology, the cumulative reward is also called return. In our setting, one trajectory can be defined as one epoch. For example, in case we have a batch size of 100 and a dataset with a size 1000, then our trajectory has 10 steps. The cumulative reward means the sum of silhouette scores computed for each batch. Our problem can be defined as a multi-armed bandit (MAB) problem which is a simplified version of *Markov decision process* ( Van Otterlo & Wiering (2012)). In MAB settings, there are no states, only actions and rewards. We keep taking actions (clustering with K-Means on the latent images) and receive rewards (silhouette score) but the state of the environment does not change.

To train the policy network, we take advantage of a simple form of policy gradient algorithm. Our policy is stochastic and we have no environment transitions. Also, we will take our return as a finite-horizon undiscounted return. Having these assumptions, we can compute the gradient in Equation 11.

When evaluating the performance of the controller, we need to consider the value for return. If it increases, then it means our model is learning.

Environment transitions and policy can be stochastic or deterministic. Assuming both are stochastic, we can formulate the probability of a trajectory $\tau$ that has $T$ steps, given action $a_t$ and policy network $\pi$ as:

$$P(\tau|\pi) = \prod_{t=0}^{T-1} P(a_t)\pi(a_t) \tag{3}$$

Then the expected return, denoted as $J(\pi)$ is:

$$J(\pi) = \int_\tau P(\tau|\pi)R(\tau) = E_{\tau \sim \pi}[R(\tau)] \tag{4}$$

Since we consider $R(\tau) = s(C)$, Equation 4 can be written as:

$$J(\pi) = \int_\tau P(\tau|\pi)s(C) = E_{\tau \sim \pi}[s(C)] \tag{5}$$

And the optimization problem in RL will be:

$$\pi^* = \underset{\pi}{argmax}(J(\pi)) \tag{6}$$

where $\pi^*$ is the optimal policy.

RL algorithms can be categorized based on what they are learning (the policy, the value function, the Q-function, or the environment model) and whether or not the agent has access to a model of the environment (access $\to$ model-based, no access $\to$ model-free). In the model-free group of algorithms, there is a major sub-group called policy optimization. Policy optimization methods rely directly upon optimizing parameterized policies with respect to the expected return (long-term cumulative reward) by gradient ascent.

To show how these algorithms update the policy network directly, we assume that our policy is a stochastic, parameterized policy denoted by $\psi_\theta$ and our return $R(\tau)$ is a finite-horizon undiscounted return. Our goal is to maximize the expected return $J(\pi_\psi) = E_{\tau \sim \pi_\theta}[R(\tau)]$. Therefore we must optimize the policy by gradient ascent (Equation 7):

$$\theta_{k+1} = \theta_k + \alpha \nabla J(\pi_\theta)|_{\theta_k}. \tag{7}$$

Algorithms that optimize the policy this way are called "Policy Gradient Algorithms" and Policy Gradient Theorem is the theoretical foundation of these algorithms. This theorem makes it possible for the gradient of the policy performance $\nabla_\theta J(\pi_\theta)$ to be numerically computed.

To be able to derive the expectation, we need to know the formula of the probability of a trajectory and a trick in calculus called the log-derivative trick. We have already shown the formula for the probability of a trajectory in Equation 3.

The log-derivative trick tells us that the derivative of $\log x$ with respect to $x$ is $\frac{1}{x}$. In our case we can use it like the following:
$$\nabla_\theta P(\tau|\theta) = P(\tau|\theta)\nabla_\theta P(\tau|\theta) \tag{8}$$

Taking those two above formulas in mind, then we can write $\log P(\tau|\theta)$ as:

$$\log P(\tau|\theta) = \sum_{t=0}^{T}(\log P(a_t) + \log \pi_\theta(a_t)) \tag{9}$$

We take the gradient of the above formula i.e. $\nabla_\theta \log P(\tau|\theta)$. Considering that the environment has no dependence on $\theta$, the gradients for $\log P(a_t)$ are zero. Therefore, Equation 9 becomes:

$$\nabla_\theta \log P(\tau|\theta) = \sum_{t=0}^{T} \nabla_\theta \log \pi_\theta(a_t) \tag{10}$$

What we get at the end is an expectation that can be estimated by sampling. Samples are acquired by letting the agent act in the environment, according to the policy $\pi_\theta$, until it reaches the terminal state, repeating this over and over and collecting a set of samples (episodes/trajectories). The gradient $\hat{g}$ would be:

$$\hat{g} = \frac{1}{|D|} \sum_{\tau \in D} \sum_{t=0}^{T} \nabla_\theta \log \pi_\theta(a_t)s(C) \tag{11}$$

Where $|D|$ is the number of samples (trajectories/episodes). Using this expression we can compute the policy gradient and update the network. This expression is the simplest version of $\nabla_\theta J(\pi_\theta)$.

## 4 EXPERIMENTS

To evaluate our method, we generate numerous datasets using the scikit-learn package with different numbers of classes ranging from 5 to 50, different numbers of samples, and different feature counts. We denote the number of features by $d$ and the length of the output vector of the controller by $n\_acts$. This means that our network predicts $k$ values in the range of $(2, n\_acts)$. In the experiments of subsection 4.2, $n\_acts$ or the actions vector has a length of 50 in all experiments. We investigated a variation of our method, where we backpropagate the gradients learned by the controller to the encoder network. However, this didn't give us any improvement. We aim to investigate this problem further.

### 4.1 IMPLEMENTATION DETAILS

The different proposed architectures for our controller are shown in Table 1. In all experiments we use Adam optimizer for the controller with learning rate $\alpha = 0.01, \beta_1 = 0.9, \beta_2 = 0.999$. For the scikit-learn experiments, we train the model for 15 epochs, with $d \in (10, 20, 30)$, $n\_acts = 50$. The generated dataset is randomly split into two parts of 0.9, 0.1 respectively for the train and test splits. For the experiments on MNIST and FMNIST (Fashion-MNIST), we train the controller for 100 epochs on the training set with $n\_acts = 20$, $d = 10$. The auto-encoder architecture used for the feature extraction is similar to the one from Xie et al. (2016) (d – 500fc – 500fc – 2000fc – 10) with $d$ being the input dimensions and 10 the size of latent dimension $z$. The autoencoder network is trained with Adam optimizer and the learning rate of $1e - 2$.

Table 1: Controller Architecture. The depth of the model increases from top to bottom. Activation function Tanh is not added for the brevity. It is applied to all the layers but the output layer. $d$ is the input feature dimension and $n\_acts$ is the length of the actions vector (range of possible $k$'s).

| Controller | Architecture |
|---|---|
| MLP1 | d-100fc-n_acts |
| MLP2 | d-100fc-100fc-n_acts |
| MLP3 | d-100fc-100fc-100fc-n_acts |

### 4.2 ABLATION STUDY

In this section, we compute the error of each experiment by subtracting the predicted value of $k$ from the ground truth value $k\prime$ and normalize it by dividing it by $k'$ ($E = \frac{\|k-k'\|}{k'}$). Then we calculate the average of all of these error values along the classes. Table 2 shows the ablation study of our method using different settings and with the mentioned error metric. The full results of this table are available in the Appendix. We increase the number of samples and the number of features in the dataset, and we see that with an increase in both of these parameters the model's performance increases. The lowest error value is achieved with MLP2 model, 10,000 samples of data, and 30 features. In all of the experiments of this section, $n\_acts$ is equal to 50.

Table 2: Results on the scikit-learn datasets generated with make_blobs function Here we show the average differences error for datasets of 5 to 50 classes.

| # of Samples | # of Features (d) | MLP1 | MLP2 | MLP3 |
|---|---|---|---|---|
| 1000 | 10 | 0.2166 | 0.2283 | 0.2166 |
| 1000 | 20 | 0.2579 | 0.2226 | 0.2129 |
| 1000 | 30 | 0.2676 | **0.1639** | 0.1934 |
| 10000 | 10 | 0.1465 | 0.1508 | 0.1353 |
| 10000 | 20 | 0.1935 | 0.1234 | 0.1266 |
| 10000 | 30 | 0.1758 | **0.1216** | 0.1380 |

### 4.3 COMPARISON WITH PREVIOUS WORKS

To compare our approach with previous work, we evaluate our method, the classical silhouette score baseline and Meta-Unsupervised-Learning from Garg & Kalai (2018) on the same synthetic dataset. For the silhouette score baseline, we calculate the silhouette score for 20 $k$ values ranging from 2 to 21 with a step size 1 and find the $k$ which gives us the highest silhouette score. For the Meta-Unsupervised-Learning method, we train a binary classification model (Multivariate linear regression with lasso) on a setting similar to the one mentioned in the original paper (instead of 339 training datasets, we train the model on 75 classification datasets downloaded from OpenML [1]) and evaluate it on the scikit-learn datasets. The datasets downloaded from the OpenML website have at most 10,000 samples, 500 features, and 10 classes, and no missing data. We apply data cleaning on each of them to make sure they include no NaN and no -inf values. The Meta-Unsupervised-Learning method learns a mapping from the silhouette score to the average random index and tries to predict the $k$ that would maximize the average random index. Based on the experiments in the ablation study we chose the model with $d = 30$, $n\_acts = 20$ and MLP2 controller. We show in Table 3 that our method is able to predict $k$ almost perfectly. Although the baseline approach achieves similar or better results, it is not feasible to use this method when the range of possible $k$'s is broad.

Table 3: Comparison to previous works. Experiment settings: # of Samples=10,000, dataset=blobs

| Ground truth | **Meta-k** | Garg & Kalai (2018) | Baseline ( Silhouette score) |
|:---:|:---:|:---:|:---:|
| 2 | **2** | 3 | 2 |
| 3 | **3** | 3 | 3 |
| 4 | **4** | 3 | 4 |
| 5 | **5** | 2 | 5 |
| 6 | **6** | 3 | 6 |
| 7 | **7** | 10 | 7 |
| 8 | 6 | 10 | **8** |
| 9 | **9** | 10 | 9 |
| 10 | **10** | 10 | 10 |

As it can be seen in Table 4, Meta-K achieves higher performance in datasets with higher dimensionality.

Table 4: Comparison to previous works on MNIST & FMNIST datasets.

| Dataset | Ground truth | **Meta-k** | Garg & Kalai (2018) | Baseline ( Silhouette score) |
|:---:|:---:|:---:|:---:|:---:|
| MNIST | 10 | **10** | 10 | 9 |
| FMNIST | 10 | 8 | **9** | 7 |

We show different clustering evaluation metric in Figure 2. It can be seen that in the MNIST and FMNIST experiments, the metrics (silhouette score, normalized mutual info, and rand index) reach their maximum at $k = 9$, $k = 7, 12$, which is sub-optimal.

---

[1]http://www.openml.org

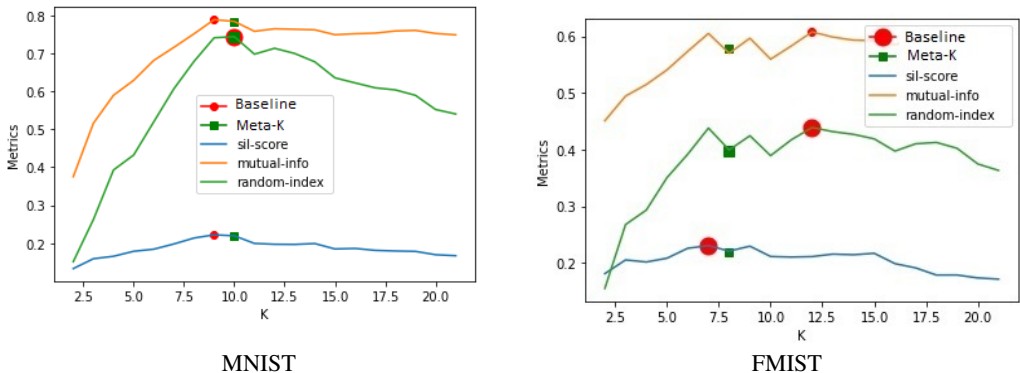

Figure 2: Results on MNIST & Fashion-MNIST datasets. The green square shows the predicted $k$ value by Meta-K and the red circle shows the maximum value on the specific metric.

## 4.4 POLICY GRADIENT OPTIMIZATION

Figure 3 shows the return function of the controller training on MNIST and FMNIST datasets. The return plot is an indication of how well the model is trained using the policy gradient optimization.

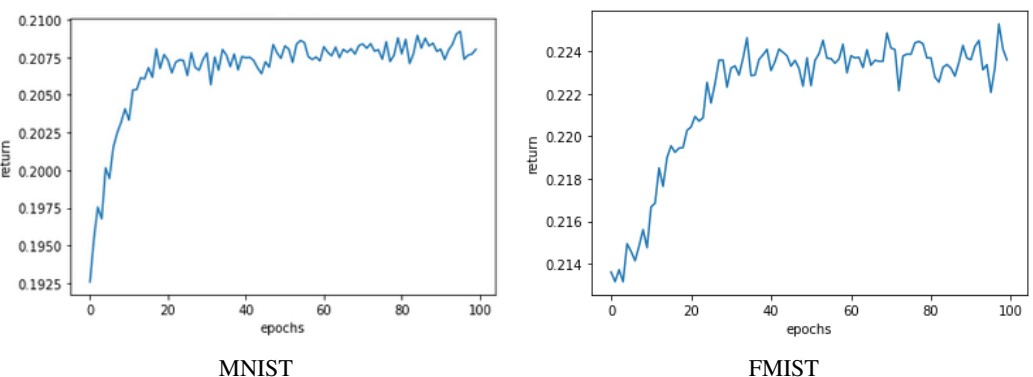

Figure 3: The graphs of cumulative return for MNIST & FMNIST datasets

## 5 CONCLUSION

In this paper, we proposed a self-supervised approach to find the optimal number of clusters for any dataset in the K-Means algorithm. We showed in our experiments that our method is able to predict the number of clusters effectively without any direct supervision. Even though our method is able to predict the number of clusters in most scenarios with distinctive features, it would be challenging if the input features are compact and not separable. Another limitation of our work is that it is dependent on the silhouette score, and the method would perform poorly if silhouette score doesn't achieve a good clustering evaluation. However our method is not limited to silhouette score and any other clustering evaluation metric can be used as well. Another point is that our method is dependent on the feature extraction network and if the encoder network is not trained properly, the controller training would also be challenging. We plan to improve our approach by using other metrics than silhouette score, adaptive layers, and Gaussian Mixture Models (GMMs) to have an end-to-end pipeline for fully automated clustering.

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

## A APPENDIX

Tables 5 - 10 show the ouptut of our method in different configurations.

Table 5: Experimentation results on the scikit-learn datasets created with make_blobs function. Variable were set to: # Samples=1000, # Features=10

| | k | | |
|---|---|---|---|
| # Classes | MLP1 | MLP2 | MLP3 |
| 5 | 6 | 3 | 2 |
| 10 | 6 | 6 | 5 |
| 15 | 6 | 21 | 14 |
| 20 | 21 | 22 | 23 |
| 25 | 24 | 24 | 23 |
| 30 | 34 | 24 | 27 |
| 35 | 34 | 34 | 35 |
| 40 | 34 | 34 | 35 |
| 45 | 34 | 34 | 34 |
| 50 | 34 | 34 | 35 |
| mean diff | 0.216635 | 0.228302 | **0.216611** |

Table 6: Experimentation results on the scikit-learn datasets created with make_blobs function. Variable were set to: # Samples=1000, # Features=20

| # Classes | k | | |
|---|---|---|---|
| | MLP1 | MLP2 | MLP3 |
| 5 | 5 | 3 | 2 |
| 10 | 15 | 3 | 13 |
| 15 | 25 | 13 | 14 |
| 20 | 25 | 21 | 24 |
| 25 | 33 | 22 | 22 |
| 30 | 33 | 33 | 33 |
| 35 | 34 | 34 | 34 |
| 40 | 34 | 34 | 34 |
| 45 | 34 | 34 | 34 |
| 50 | 34 | 35 | 34 |
| mean diff | 0.257968 | 0.222635 | **0.212968** |

Table 7: Experimentation results on the scikit-learn datasets created with make_blobs function. Variable were set to: #Samples=1000, #Features=30

| # Classes | k | | |
|---|---|---|---|
| | MLP1 | MLP2 | MLP3 |
| 5 | 3 | 5 | 2 |
| 10 | 13 | 6 | 12 |
| 15 | 25 | 17 | 14 |
| 20 | 24 | 17 | 21 |
| 25 | 22 | 23 | 24 |
| 30 | 22 | 34 | 35 |
| 35 | 34 | 34 | 31 |
| 40 | 34 | 34 | 33 |
| 45 | 34 | 34 | 35 |
| 50 | 35 | 34 | 35 |
| mean diff | 0.267635 | **0.163968** | 0.193484 |

As expected, by increasing the size of the datasets, our model's predictions became more accurate. However, even with a dataset of size 1000, our model's performance was acceptable when classes were of size 5 to 35. A change in the number of features (space dimensionality) did not make a visible difference here.

Table 8: Experimentation results on the scikit-learn datasets created with make_blobs function. Variable were set to: #Samples=10,000, #Features=10

| # Classes | k MLP1 | MLP2 | MLP3 |
|---|---|---|---|
| 5 | 5 | 5 | 5 |
| 10 | 12 | 9 | 11 |
| 15 | 16 | 14 | 12 |
| 20 | 18 | 19 | 21 |
| 25 | 19 | 23 | 22 |
| 30 | 22 | 25 | 26 |
| 35 | 34 | 27 | 32 |
| 40 | 34 | 27 | 36 |
| 45 | 39 | 31 | 34 |
| 50 | 36 | 41 | 34 |
| mean diff | 0.146524 | 0.150802 | **0.135349** |

Table 9: Experimentation results on the scikit-learn datasets created with make_blobs function. Variable were set to: #Samples=10,000, #Features=20

| # Classes | k MLP1 | MLP2 | MLP3 |
|---|---|---|---|
| 5 | 3 | 5 | 6 |
| 10 | 6 | 9 | 8 |
| 15 | 16 | 14 | 16 |
| 20 | 16 | 19 | 19 |
| 25 | 20 | 20 | 23 |
| 30 | 23 | 26 | 30 |
| 35 | 33 | 29 | 33 |
| 40 | 39 | 37 | 33 |
| 45 | 39 | 37 | 37 |
| 50 | 39 | 37 | 37 |
| mean diff | 0.193548 | **0.123421** | 0.126659 |

Table 10: Experimentation results on the scikit-learn datasets created with make_blobs function. Variable were set to: #Samples=$10,000$, #Features=30

| # Classes | k MLP1 | MLP2 | MLP3 |
|---|---|---|---|
| 5 | 4 | 5 | 4 |
| 10 | 6 | 8 | 11 |
| 15 | 16 | 14 | 14 |
| 20 | 18 | 20 | 17 |
| 25 | 22 | 23 | 20 |
| 30 | 31 | 21 | 26 |
| 35 | 33 | 37 | 34 |
| 40 | 33 | 37 | 36 |
| 45 | 33 | 37 | 35 |
| 50 | 33 | 37 | 41 |
| mean diff | 0.175881 | **0.121659** | 0.138079 |

