# OpenReview forum: "Meta-k: Towards Unsupervised Prediction of Number of Clusters"
_ICLR.cc/2021/Conference — Reject_

### Official Review · AnonReviewer3 · 2020-10-25
**Review for Meta-k: Towards Unsupervised Prediction of Number of Clusters**

**Rating:** 3
**Confidence:** 5

**Review:**

The reviewed paper presents a completely unsupervised framework Meta-K for predicting the number of clusters. The approach advocated in the paper comprises two main parts: autoencoder for feature extraction and multilayer perceptron (MLP) for predicting the number of clusters. Autoencoder is used if necessary to decrease the dimensionality of the input data. The MLP is trained using policy gradient optimization schema to predict the best (according to silhouette score) number of clusters k in the given dataset. Overall, the authors show that their approach achieves near-optimum results on both a number of synthetic datasets as well as on two well-known computer vision datasets: MNIST and FMNIST.

Strong points:
* Overall, the paper is well written. The reading flow is smooth and clear without major disturbances. Alike text, figures (especially Figure 1), greatly facilitates the understanding of the paper.
* The review of the related literature seems to be very thorough and lists most of the relevant papers in the field. This section helps to put the current work into context and explain the limitations of approaches proposed by past research.

Weak points as well as reasons for the score:
* The main problem with the paper seems to be the apparent discrepancy between the performance and complexity of the proposed approach (Meta-k), and the baseline method (silhouette score). While both methods are completely unsupervised, the former is a complex composite pipeline, which includes two neural networks, one of which is trained using a reinforcement learning framework, the latter is a simple formula that can be calculated for as many clusters as necessary in a relatively short time. Both methods, according to the authors, perform on par, with the baseline being marginally better on synthetic data. Moreover, according to the paper, the Meta-k framework is trained to find the best number of clusters by maximizing the silhouette score in the first place. Unfortunately, the authors largely omitted a question of why their approach should be preferred over the baseline. Only once, when commenting on results of the comparison with baseline, the authors noted: "although the baseline approach achieves similar or better results, it is not feasible to use this method when the range of possible k's is broad", not disclosing details as to what exactly about a broad range of k's makes the baseline method infeasible, and why Meta-k must be preferred in such circumstances. Overall, the benefit of using a more complex method, such as Meta-k should be clearly stated in the paper and also extensively experimentally verified.

* The lack of experimentation is the second most important concern and also the reason for the current score. Meta-k has been used on several datasets, including a number of synthetic datasets generated using sklearn package and also well-known MNIST and FMNIST. While experiments on synthetic data is a great way to confirm the initial hypothesis about the model, superior performance on real-world data is what can be of interest to the community. Although, MNIST and FMNIST are good starting points, a lot more datasets exist where the number of clusters (i.e. classes) is known: e.g. CIFAR-10, CIFAR-100, ImageNet, etc.
The same can be said about the competitors, as currently only one approach (except baseline) is compared to Meta-k.

* Also, it is not made clear to the readers why policy gradient optimization was used for training the controller network, rather than for example common SGD with custom loss based on silhouette score. The RL training in such circumstances and without proper explanations seems to be an excessive measure.

* The paper can be better structured. Section 3, might have been called "Methods", where the authors could have described not only their proposed solution but also the competitor approache(s) that they decided to compare with. It might be a good idea to move some part of the policy gradient optimization discussion to supplementary materials, leaving more room for experiments and results. Section "Experiments" might have been split into "Experiments" and "Results", with "Experiments" focusing on performed experiments and their setup, while "Results" remaining more focused on obtained outcomes and relevant interpretation.

* Large parts of the introduction are overlapping or repeated in related work, it is not necessary to discuss relevant literature in the introduction, this part should be moved to related work completely.

* Over a number of occasions (in the abstract, introduction, and related work) the authors point out a close relationship between their approach and meta-learning or 'learning to learn' concept, hence the name of the approach (Meta-k). However, nowhere in the paper, the authors seem to clarify which part of their method performs meta-learning. While it might be understood that the authors refer to the fact that their approach trains the controller network using the unsupervised signal from the silhouette score. It is however might be a good idea to make this connection explicit.

Minor comments:

* Training curves for the policy gradient optimization presented in Section 4.4 are left with no interpretation, and to the reader it remains unclear why they were presented in the final section in the first place.

* Caption of Figure 1 should ideally also explain what x, x', z, and other variables mean.

* The third item on the list of contributions, interprets experimental results and can hardly be considered as a contribution of the paper.

* Some of the terms related to reinforcement learning, such as environment transitions or finite-horizon undiscounted return must be explained before use.

* Figure 2, could have been made more clear by adding vertical lines that would correspond to the true value of the number of clusters.

* All acronyms, such as MLP should be defined before being used.

---

### Official Review · AnonReviewer2 · 2020-10-28
**The number of clusters is optimized with respect  to the silhouette score in k-means clustering based on policy gradients.**

**Rating:** 4
**Confidence:** 3

**Review:**

The paper uses policy gradients in a bandit setting to learn the optimal number of clusters, k,  in k-means clustering based on the silhouette score. Finding k that leads to the highest silhouette score is a more specific problem that what the paper title promises. The approach is well-described and supported by experiments on simulated and real-world data.
Phrasing the task at hand as hyper-parameter optimization problem, it could be tackled with a variety of well known tools, e.g. Gaussian Process based HPO, grid search, random sampling, etc. The authors need to compare the policy gradient method to these existing approaches.

Detailed comments:
- k values for competitors are not chosen properly
- clearly written and presented
- is the generation of synthetic data noise-free?
- what is the y-axis in Fig.2 ?
- "the phenomenon is called the curse of dimensionality" -> maybe better: "the underlying phenomenon"
- eq. (6): argmax should be formatted as \text{argmax).
- Please check spacing and parentheses around citations.
- "our approach can predict the number of clusters in most settings with an insignificant error." -> "insignificant" is vacuous, here.
- eq. 8 is not correct

---

### Official Review · AnonReviewer1 · 2020-11-02

**Rating:** 4
**Confidence:** 4

**Review:**

The paper proposes a method to determine the number of clusters k to use with k-means. The method uses a of policy gradient inspired method without states to predict the number of clusters to use.

The idea of the paper and the presented results are promising and indicate that the idea has merit. However, there are several concerns with the paper as it stands. The underlying assumption of the paper appears to be that there is a single correct value of clusters is correct and that the used metric is capable of reliably telling good from bad clustering solutions apart.

Another aspect that is entirely ignored in the discussion and description of the method is that the training of the policy is performed in batches. This means that unless the the data in each batch is well distributed over all classes present in the data the metric would seem to be different for different batches leading to an optimization problem with a moving target. From the text it is not clear to me how this challenge is addressed by the proposed method. In practice it is impossible to guarantee either balancedness or stratification of the data presented to an unsupervised algorithm so it would seem unreasonable to require this. This is a crucial challenge for the proposed method which is sadly not addressed at all.

From a related work point of view the paper focuses only on k-means and ignores other clustering methods which determine the number of clusters on their own, e.g. Affinity Propagation or LDA. While k-means and spectral clustering are widely used it would be good to include and compare to other methods that already are capable of determining the number of clusters directly from data. Furthermore, the paper does not speak to or compare with the well established methods to determine k for k-means such as the BIC or AIC.

Another aspect that the experiments don't really make clear is what the advantage of the proposed method is compared to simply increasing k and using the proposed metric for each value and picking the best k as was done for the baseline method. Obviously there is a cost involved in running k-means repeatedly but training the policy is not free either and guaranteeing convergence of a deep network to a reasonable solution would seem more challenging than just computing the metric value repeatedly. A discussion of this would be greatly appreciated.

The description of the method is at times a bit hard to follow. This is partly due to the paper intermixing description of the method and background information. An example of this is 3.1 where RL without with some variation on MDP is introduced intermingled with the actual setup used for the proposed method. It would have been easier to read if first the entire MDP and bandit setting was presented before describing how this is used in the proposed method. Certain aspects also never become clear. It appears that the network outputs a high-dimensional feature vector which is used to determine k. Figure 1 makes this output appear to be a scalar when the text makes it sound like a vector or distribution. Overall this adds to the challenge that as it stands it would be hard to implement the proposed method, the general idea is conveyed but many of the nuanced details remain unclear.

---

### Decision · Program_Chairs · 2021-01-07
**Final Decision**

**Decision:**

Reject

**Comment:**

The reviewers were unanimous that this submission is not ready for publication at ICLR. Concerns were raised about clarity of the exposition, as well as lack of sufficient experiments comparing to related work.